# Intrapartum Factors Affecting Abnormal Lipid Profiles in Early Postpartum Period

**DOI:** 10.3390/jpm13030444

**Published:** 2023-02-28

**Authors:** Da Kyung Hong, Hee Young Cho, Ji Youn Kim, Hee Jin Park, Dong Hyun Cha, Sung Shin Shim, Bo Seong Yun

**Affiliations:** 1Department of Obstetrics and Gynecology, CHA Gangnam Medical Center, CHA University, Seoul 06125, Republic of Korea; 2Department of Obstetrics and Gynecology, Seoul National University College of Medicine, Seoul 03080, Republic of Korea; 3Department of Obstetrics and Gynecology, CHA Ilsan Medical Center, CHA University, Goyang-Si 10414, Republic of Korea

**Keywords:** cholesterol, lipid profiles, lipoprotein, postpartum, pregnancy

## Abstract

The aim of this research is to investigate the risk factors during pregnancy affect abnormal lipid profiles in women with early postpartum period. This was a single-center retrospective study including 869 women who delivered between December 2017 and May 2019. We collected total cholesterol levels, both at 24–28 GWs and 1 month before delivery. Lipid profiles such as total cholesterol, high-density lipoprotein (HDL), triglyceride (TG) and low-density lipoprotein (LDL) at 6 weeks after delivery were retrieved. Subjects were categorized into 3 groups such as normal, borderline and abnormal group according to the lipid profile levels. The risk factors associated with borderline to abnormal HDL level were body mass index (BMI) of pre-pregnancy (OR = 1.182, 95% CI: 1.116–1.252, *p* < 0.001), weight gain during pregnancy (OR = 1.085, 95% CI: 1.042–1.131, *p* < 0.001) and hypertension (HTN) (OR = 3.451, 95% CI: 1.224–9.727, *p* = 0.02). The risk factors associated with borderline or abnormal TG were BMI of pre-pregnancy, weight gain during pregnancy and weight reduction after delivery. HTN was associated with borderline to abnormal TG in postpartum (OR = 2.891, 95% CI: 1.168–7.156, *p* = 0.02), while GDM correlated purely with abnormal TG in postpartum (OR = 2.453, 95% CI: 1.068–5.630, *p* = 0.03). Abnormal lipid profiles in postpartum were significantly associated with BMI of pre-pregnancy, weight gain during pregnancy and weight reduction after delivery. In addition, pregnancy-related HTN was highly associated with abnormal HDL level, and GDM was associated with abnormal TG level in the early postpartum period.

## 1. Introduction

Cholesterol and other lipid profiles increase during pregnancy physiologically towards term and then decrease after delivery. In the late second trimester, increased amounts of free fatty acids are released into the systemic circulation, affected by maternal hormonal changes such as increased insulin resistance, progesterone, 17-ß estradiol and human placental lactogen [1]. These accumulated lipids in late pregnancy naturally might provide a reservoir of fatty acids for fetal growth and placental tissue steroid synthesis.

Many studies have described that intrapartum hyperlipidemia is related to body mass index (BMI), maternal weight gain, maternal nutrition, and pre-pregnancy lipid levels [1]. In addition, various complications of pregnancy such as preeclampsia (PE), gestational diabetes mellitus (GDM), and preterm birth have significant effects on lipid metabolism and plasma lipid level during pregnancy [2,3,4]. It is well established that abnormal levels of lipid early in pregnancy are associated with an increasing risk of PE [2].

Recently, it is reported that the lipid profile from the third trimesters is a possible better prediction marker for maternal and fetal outcome [3]. In a systemic review and meta-analysis published in 2015, triglycerides (TG) were significantly elevated among women with GDM, compared to women without insulin resistance throughout pregnancy periods [4]. Also, dyslipidemia is often observed in preterm birth (PTB), which is considered as a vascular pathologic abnormality [5]. Previous study reported that risk of PTB increased with high TG level but did not associate with HDL and LDL cholesterol [5]. Elevated plasma lipid and lipoproteins may induce endothelial dysfunction secondary to oxidative stress [6], and might be independent risk factors for vascular disease. Increased peroxidation of these elevated plasma lipids causes oxidative stress by free radicals and lipid peroxides. Lipid peroxides are toxic compounds and damage endothelial cells [7].

The placentas of obese women at term contained increased total lipid level and an accumulation of proinflammatory mediators and macrophages compared with normal weight women [8,9]. The placental transcriptome of obese women with GDM shows activation of genes related to lipid metabolism at a molecular level. Obese women have an increased risk of a failed trial of labor, cesarean delivery, and endometritis. Maternal BMI is inversely proportional the length of labor in nulliparous women [10]. In the postpartum period, maternal obesity is a possible risk factor for venous thromboembolism and a higher risk of pulmonary embolism [11].

Large prospective studies on adults have clearly shown that high low-density lipoprotein (LDL) and low high-density lipoprotein (HDL) levels are significantly associated with cardio-vascular risk [12]. In pregnant women, elevated lipid profiles can mostly return to pre-pregnancy level after natural delivery. However, if abnormal lipid profiles sustain after delivery, these may increase the risk of long-term cardiovascular complications in later life. Furthermore, in most cases, lipid profiles are not routinely checked after delivery. 

Therefore, we identified the range of lipid profiles during pregnancy and early postpartum period, and investigated which risk factors during pregnancy affect borderline or abnormal lipid profiles including total cholesterol, HDL, TG and LDL in women with early postpartum period. The aim of this study is to reveal the levels of lipid profiles of women before and after delivery, and provide the appropriate care to prevent possible cardiovascular risk.

## 2. Materials and Methods

The subjects of this retrospective cohort study were women who delivered at CHA Gangnam Medical Center, CHA University and who received the Postpartum Health Service between December 2017 and May 2019. A total of 1,065 women were selected during the study period. Women who did not have information of total cholesterol levels during pregnancy and who delivered at less than 28 gestational weeks were excluded. Ultimately, 869 women were included for this study (Figure 1). The Postpartum Health Service in CHA Gangnam Medical Center is a package program to estimate the physical and mental health of women at 6 weeks after delivery and serve as a guide to treat women who have been discharged. The Postpartum Health Service also includes lipid profiles such as fasting total cholesterol, HDL, TG and LDL levels. To identify the range of total cholesterol during pregnancy, women without information of total cholesterol during pregnancy were excluded. This study was conducted after approval from the Institutional Review Board (IRB) of CHA Gangnam Medical Center, CHA University (2021-03-007-001, approved 04/21/2021). Informed consent was waived due to the retrospective nature of the study by IRB of the CHA Gangnam Medical Center, CHA University.

With review of the medical records, we retrospectively collected information including age, parity, BMI of pre-pregnancy, weight gain during pregnancy (weight of pre-pregnancy subtracted from weight prior to delivery), weight loss after delivery (weight postpartum at 6 weeks subtracted from weight prior to delivery), the presence of in-vitro fertilization (IVF) during current pregnancy, history of hypothyroidism or hyperthyroidism before pregnancy, twin pregnancy at delivery time, the presence of preterm birth (delivered less than 37 gestational weeks in the current pregnancy), hypertensive disease (gestational hypertension: HTN, PE or PE with chronic HTN), diabetes mellitus type 2 (gestational diabetes mellitus: GDM or diabetes mellitus: DM), and thyroid disease diagnosed during pregnancy [3,5,7]. Pregnant women who had taken hyperthyroid or hypothyroidmedication but maintaining euthyroidismwere included in this study, regardless of diagnosing hypothyroidism prior to or during pregnancy. 

In addition, we also collected total cholesterol levels, both at 24–28 gestational weeks (GWs) in the 2nd trimester and 1 month before delivery in the 3rd trimester. Also, we collected lipid profiles (total cholesterol, HDL, TG, and LDL) at 6 weeks after delivery which were included in Postpartum Health Service program. LDL was calculated by Friedewald formula: LDL (mg/dL) = total cholesterol (mg/dL) − HDL (mg/dL) − TG (mg/dL)/5 [13]. We categorized 3 groups for lipid profiles level into normal, borderline, and abnormal groups. The range of lipid profiles was defined as below; For total cholesterol, normal (<200 mg/dL), borderline (200–239 mg/dL), and abnormal (≥240 mg/dL); For HDL, normal (>60 mg/dL), borderline (39–60 mg/dL), and abnormal (≤40 mg/dL); For TG, normal (<150 mg/dL), borderline (150–199 mg/dL), and abnormal (≥200 mg/dL); For LDL, normal (<130 mg/dL), borderline (130–159 mg/dL), and abnormal (≥160 mg/dL) [14].

Statistical analysis were performed using the IBM SPSS ver. 26.0 (IBM Corp., Armonk, NY, USA). Numerical data were shown as mean ± SD, and categorical data were described with numbers and percentages. To compare the continuous variables among three groups, ANOVA was used and for post hoc analysis, least significant difference was used. Dichotomous variables were compared using chi-square test. Multiple logistic regression analysis was performed to evaluate independent risk factors between normal group and borderline plus abnormal groups, and between normal plus borderline groups and abnormal group, respectively. A value of *p* < 0.05 was considered an indicator of statistical significance.

## 3. Results

The clinical characteristics of the study population are on Table 1. The mean level of total cholesterol in the 2nd and 3rd trimester were 242.7 ± 37.0 mg/dL and 270.2 ± 45.3 mg/dL respectively, while the mean level of total cholesterol in postpartum was 208.1 ± 34.4 mg/dL. The mean level of HDL, TG and LDL in postpartum was 59.7 ± 12.8 mg/dL, 100.7 ± 66.1 mg/dL and 128.2 ± 30.5 mg/dL respectively. Total cholesterol level of postpartum was positively correlated with total cholesterol level of the 2nd trimester (r = 0.552, *p* < 0.001) and 3rd trimester (r = 0.519, *p* < 0.001). In addition, a positive correlation of total cholesterol level between 2nd and 3rd trimester was also observed (r = 0.801, *p* < 0.001). 

Scatter plots of the correlations of total cholesterol levels per period are depicted in Figure 2.

We evaluated the associated factors in clinical findings with borderline or abnormal group of lipid profiles in postpartum women in Table 2. In addition, we also performed multivariate analyses to predict independent risk factors associated with abnormal lipid profiles during pregnancy and postpartum. There were no association between clinical findings and borderline or abnormal level of total cholesterol in postpartum.

When assessing HDL level in postpartum by multivariate analysis, independent risk factors associated with borderline to abnormal HDL were BMI of pre-pregnancy (OR = 1.182, 95% CI: 1.116–1.252, *p* < 0.001), weight gain during pregnancy (OR = 1.085, 95% CI: 1.042–1.131, *p* < 0.001) and HTN (OR = 3.451, 95% CI: 1.224–9.727, *p* = 0.02) (Table 3). Considering the purely abnormal HDL level in postpartum, risk factors included age (OR = 0.884, 95% CI: 0.786–0.994, *p* = 0.04), BMI of pre-pregnancy (OR = 1.157, 95% CI: 1.023–1.307, *p* = 0.02), weight gain during pregnancy (OR = 1.139, 95% CI: 1.039–1.249, *p* = 0.006) and HTN (OR = 9.945, 95% CI: 2.793–35.416, *p* < 0.001). 

In Table 4, when assessing TG level in postpartum, BMI of pre-pregnancy (OR = 1.178, 95% CI: 1.106–1.254, *p* < 0.001), weight gain during pregnancy (OR = 1.108, 95% CI: 1.054–1.165, *p* < 0.001), weight reduction after delivery (OR = 0.791, 95% CI: 0.727–0.861, *p* < 0.001) and HTN (OR = 2.891, 95% CI: 1.168–7.156, *p* = 0.022) were independent risk factors associated with borderline to abnormal TG in postpartum by multivariate analysis. Whereas, the risk factors with purely abnormal TG were BMI of pre-pregnancy (OR = 1.148, 95% CI: 1.059–1.243, *p* < 0.001), weight reduction after delivery (OR = 0.874, 95% CI: 0.783–0.976, *p* = 0.017) and GDM (OR = 2.453, 95% CI: 1.068–5.630, *p* = 0.034). 

In Table 5, however, considering LDL level in postpartum by multivariate analysis, there were no risk factors in clinical findings that were associated with borderline or abnormal level of LDL in postpartum.

## 4. Discussion

This study demonstrated that weight related factors and pregnancy-related HTN or GDM were highly associated with borderline or abnormal HDL and TG in postpartum, whereas there were no risk factors associated with borderline or abnormal total cholesterol and LDL. Higher BMI of pre-pregnancy, higher weight gain during pregnancy and pregnancy-related HTN were associated with abnormal HDL including borderline zone. On the other hand, abnormal TG in postpartum were associated with higher BMI of pre-pregnancy and less weight reduction after delivery. In addition, pregnancy-related HTN had an effect on borderline to abnormal TG, while GDM was highly associated with purely abnormal level of TG in postpartum. These results imply that women who were originally obese, and gained more weight during pregnancy, or lost less weight after delivery would perform better lipid profile levels, because they have the risk of borderline to abnormal profiles which may be associated with cardiovascular risk if abnormal lipid profile levels persist for a long time. Especially, with complicated pregnancy such as pregnancy-related HTN or GDM, HDL or TG should be more specifically identified because total cholesterol or LDL was not a representative profile.

Hyperlipidemia is one of the important risk factors for developing coronary heart disease (CHD), which is still a leading cause of death in the USA and Europe [15]. Prospective population studies announced LDL concentrations as a positive predictor of CHD and stroke [16], and also found that HDL level proved to be a potent cardio-protective factor [17]. In addition, hypertriglyceridemia also develops the formation of atherogenic LDL cholesterol and reduces cholesterol clearance in circulation by decreasing HDL-C mediated transport to the liver [18]. However, this study showed that only predictive factors during pregnancy are associated with abnormal HDL and TG in postpartum, not total cholesterol or LDL. This means that the change of lipid profiles appear in various ways after delivery, but some factors or complicated pregnancy were closely related with only HDL or TG, considering pregnancy status. 

Total cholesterol levels were gradually elevated up to borderline (242.7 ± 37.0 mg/dL) to high level (270.2 ± 45.3 mg/dL) as pregnancy progresses, and returned to almost normal level after delivery (208.1 ± 34.4 mg/dL) in our study. This is similar to other literatures which showed that total cholesterol generally increases during pregnancy by approximately 40% and returns to pre-pregnancy levels within one year postpartum [1]. Most of the women in our study showed the correlation between the level of total cholesterol during pregnancy and postpartum. Higher total cholesterol in the 2nd trimester or 3rd trimester were the same in postpartum. Accumulated lipids in late pregnancy have an important role to provide a reservoir of fatty acids for fetal growth and placental tissue steroid synthesis. There is an anabolic phase with an increase in lipid synthesis and fat storage in preparation for the increases in fetal energy needs in late pregnancy [19]. 

Obesity is associated with increased oxidative stress and inflammation, which has an impact on cardiovascular disease [20]. In most of obese patients, glucose, insulin, total cholesterol, HDL, and LDL are typically elevated. Women with obesity before pregnancy should be offered interventions aimed at assisting long-term weight loss, as recommended by the US Preventive Task Force, because numerous complications associated with pregnancy can occur [21]. Maternal obesity contributes to increased fetal birth weight and higher BMI in adolescent period. Epigenetics are thought mediate these effects and DNA methylation changes associated with maternal obesity. Prenatal malnutrition associated with less DNA methylation at the imprinted *IGF2* gene locus [22]. A variety of epigenetic marks predominantly interfering with placental development, function, and metabolism were found to be potentially associated with fetal growth restriction [23]. Among epigenetic factors, *IGF2*/*H19* genes are strong candidates for birth weight variation and IGF2 serum level and mRNA expression level have positively correlated with fetal birth weight [24].

During pregnancy, fat is accumulated in both peripheral and central sites. However, after delivery, the accumulation of central adipose tissue becomes a mainstream representing visceral fat, because peripheral fat is mobilized [25]. Furthermore, in a prospective cohort study of Mexican women, weight retention or gain in the first year postpartum was highly correlated with obesity, insulin resistance, high TG and low HDL at even 6 years, implicating their long-term risk of obesity and cardiovascular disease [26]. In our results, higher BMI of pre-pregnancy showed strong associations with abnormal HDL and TG levels in postpartum. Interestingly, regardless of BMI, heavy weight gain during pregnancy has an impact on abnormal HDL and mild elevation of TG. Weight retention even after delivery may be an important risk factor for abnormal TG in postpartum. Due to the fact that body weight is considered as a correctable pre-pregnancy factor, women with high BMI in pre-pregnancy should be advised to recommend weight loss during pregnancy preparation.

In our study, postpartum HDL level showed a high association with HTN. In terms of intrapartum HDL level in PE, several studies have reported that HDL level decreases more than non-PE. One of those studies presented its mechanism as potentially increased diameter of HDL and reduced antioxidant activity of HDL during pregnancy in PE women, concerning the lifetime cardiovascular risk associated with those changes [27,28]. Usually after delivery, HTN or proteinuria in women with PE returns to normal. However, our study showed that women with pregnancy-related HTN still had 3.4 times of less than 60 mg/dL of HDL and 9.9 times of less than 40 mg/dL of HDL in early postpartum period after correcting all other confounding factors. This is consistent with the need for continuous evaluation of HDL after delivery in women of pregnancy-related HTN. Abnormal HDL was inversely associated with age, and it was the general result of non-pregnant adults. When HDL was evaluated with an 10-year-age interval in adults, older age correlated with more decreased HDL levels [29,30]. However, our study investigated the specific cohort of pregnant women, resulting in a small difference of age interval between women. Therefore, age factor in this result might not have a significant discrimination. In addition, there was a big difference of sample size between normal plus borderline groups and abnormal group, which showed weaker statistical significance than such of similar sample size. In the future, further prospective studies with similar sample size would be needed to draw a more reasonable correlation.

During pregnancy, the association between GDM and maternal dyslipidemia is well established. In a systematic review, high TG levels and low HDL levels were observed in women with GDM [4]. Regarding postpartum lipid levels of GDM women, several studies reported that GDM women still have dyslipidemia [31,32]. One study showed that only HDL and TG level differed significantly at 2 and 12 months after delivery between GDM and non-GDM group [31]. Another study reported that the overall prevalence of postpartum dyslipidemia was 52%; higher cholesterol in 44%, higher LDL in 33% and higher TG in 16% at 6 weeks postnatally in GDM women [32]. In our study, GDM was independently associated with abnormal TG level, more than 200mg/dL, at 6-weeks postpartum. Women with GDM are known to have a 5-fold increase in the risk of DM in their lifetime than women without GDM, but even worse, unexpectedly remaining hypertriglyceridemia may adversely affect future DM and cardiovascular development. Low birthweight is a potential marker for poor fetal nutritional status and has been related with metabolic abnormalities, such as cardiovascular disease and type 2 diabetes in adulthood [22].

Our study confirmed that weight-related factors such as BMI of pre-pregnancy, weight gain during pregnancy and weight reduction after delivery, showed abnormal lipid profiles at postpartum. We also suggested the possibility that pregnancy-related HTN and GDM, which are widely known to increase female cardiovascular risk, may also be the additional predictable factors associated with lipid profile abnormalities at postpartum. Furthermore, we have identified lipid types that are specific to each of these factors. Although the implication of factor-specific elevated lipid type deserves further exploration of their role in long term cardiovascular risk in women, our result helps clinically to determine which type of lipid should be carefully focused in each of the factors identified during maternal care. Women with these factors should be screened for abnormal postpartum lipid profile, and further follow-up is essential for normalization.

The limitations of this study were that this study was a retrospective study, and no information about Omega-3 was taken during pregnancy which affects lipid profiles. In addition, the lipid profiles at early pregnancy or other lipid profiles except for total cholesterol were not obtained, due to lack of understanding the serial changes of lipid profiles over pregnancy. Although our postpartum data were collected at 6 week after delivery based on the definition of postpartum, long term follow up data would be needed to evaluate more accurate cardiovascular risk of future lifetime.

Abnormal lipid profiles in postpartum were significantly associated with weight related factors, such as BMI of pre-pregnancy, weight gain during pregnancy and weight reduction after delivery, and complicated pregnancy including pregnancy-related HTN and GDM. Specifically, pregnancy-related HTN was highly associated with abnormal HDL level, and GDM was associated with abnormal TG level in early postpartum period. In order to decrease the risk of cardiovascular disease, efforts should be made to correct modifiable factors such as controlling weight before pregnancy, avoiding excessive weight gain during pregnancy, and trying weight reduction after delivery. In addition, it is recommended that HDL and TG level be evaluated in women with HTN and GDM. However, more long-term studies to directly speculate cardiovascular risk are warranted to establish stronger consequences.

## Figures and Tables

**Figure 1 jpm-13-00444-f001:**
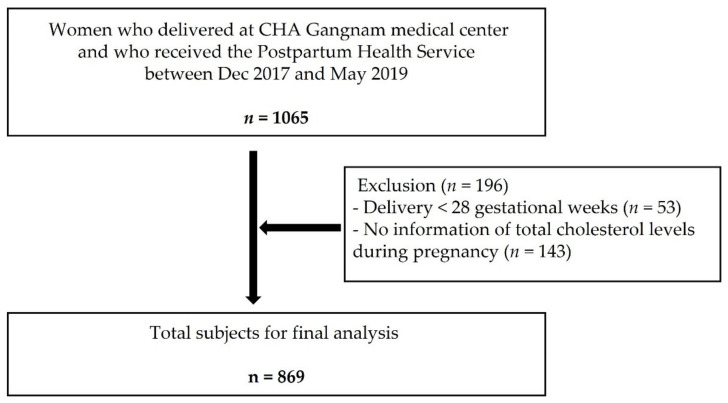
Schematic presentation of study with exclusion criteria.

**Figure 2 jpm-13-00444-f002:**
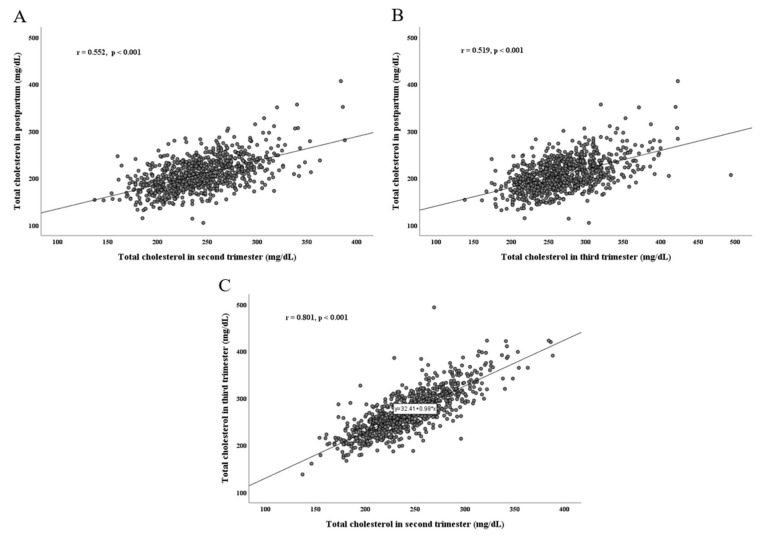
The correlation of total cholesterol level per pregnancy period. Total cholesterol level of postpartum was positively correlated with total cholesterol level of 2nd trimester (r = 0.552, *p* < 0.001) (**A**) and 3rd trimester (r = 0.519, *p* < 0.001) (**B**). The positive correlation of total cholesterol level between 2nd and 3rd trimester was also observed (r = 0.801, *p* < 0.001) (**C**).

**Table 1 jpm-13-00444-t001:** Baseline characteristics of the subjects.

Characteristics		*n* = 869
Age	<35 yrs	418 (48.1)
	≥35 yrs	451 (51.9)
Parity	Nulliparous	674 (77.6)
	Mutiparous	195 (22.4)
Thyroid disease history	Hypothyroidism	45 (5.2)
	Hyperthyroidism	9 (1.0%)
IVF	No	598 (68.8)
	Yes	271 (31.2)
BMI (kg/m^2^)	Pre-pregnancy	21.0 ± 2.9
	Pre-delivery	26.1 ± 3.2
	Postpartum 6 weeks	22.5 ± 3.0
Weight gain (kg)	During pregnancy	13.5 ± 4.6
Weight reduction (kg)	After delivery	9.4 ± 2.9
Twin pregnancy at delivery		40 (4.6)
Preterm birth		76 (8.7)
HTN	Gestational HTN	7 (0.8)
	Preeclampsia	20 (2.3)
	Preeclampsia with chronic HTN	2 (0.2)
DM	GDM	55 (6.3)
	DM	2 (0.2)
Thyroid disease after pregnancy	Hypothyroidism	69 (7.9)
	Hyperthyroidism	9 (1.0)
Total cholesterol (mg/dL)	2nd trimester	242.7 ± 37.0
	3rd trimester	270.2 ± 45.3
	Postpartum	208.1 ± 34.4
HDL (mg/dL)	Postpartum	59.7 ± 12.8
TG (mg/dL)	Postpartum	100.7 ± 66.1
LDL (mg/dL)	Postpartum	128.2 ± 30.5

IVF: in vitro fertilization, BMI: body mass index, HTN: hypertension, DM: diabetes mellitus, GDM: gestational diabetes mellitus, HDL: high-density lipoprotein, TG: triglyceride. data was shown as mean± SD or number (%).

**Table 2 jpm-13-00444-t002:** Factors associated with abnormal total cholesterol in postpartum.

	Normal	Borderline	Abnormal		Normal vs. Borderline to Abnormal			Normal to Borderline vs. Abnormal		
	(*n* = 372)	(*n* = 360)	(*n* = 137)	*p*	OR	CI	*p*	OR	CI	*p*
Age (years)	34.4 ± 3.9	34.8 ± 3.8	35.3 ± 4.0	0.057	1.036	0.997–1.076	0.072	1.043	0.990–1.099	0.111
Multi-parity	83 (22.3)	80 (22.2)	32 (23.4)	0.961	0.927	0.659–1.303	0.661	0.942	0.598–1.484	0.796
IVF	114 (30.6)	111 (30.8)	46 (33.6)	0.510	0.921	0.663–1.281	0.626	1.049	0.679–1.621	0.829
BMI (kg/m^2^)	20.9 ± 2.9	20.8 ± 2.9	21.3 ± 3.0	0.275	0.999	0.950–1.050	0.961	1.048	0.984–1.116	0.146
Weight gain (kg)	13.6 ± 4.7	13.5 ± 4.8	13.1 ± 4.0	0.498	1.008	0.971–1.047	0.667	0.993	0.944–1.044	0.779
Weigh reduction (kg)	9.6 ± 3.0	9.4 ± 2.9	9.1 ± 2.6	0.224	0.960	0.903–1.021	0.197	0.964	0.888–1.046	0.375
Twin pregnancy	16 (4.3)	19 (5.3)	5 (3.6)	0.693	1.212	0.565–2.597	0.621	0.978	0.325–2.946	0.969
Preterm birth	30 (8.1)	37 (10.3)	9 (6.6)	0.352	1.081	0.624–1.873	0.781	0.713	0.320–1.588	0.408
HTN	13 (3.5)	15 (4.2)	1 (0.7)	0.159	0.926	0.416–2.061	0.850	0.183	0.024–1.397	0.102
GDM	20 (5.4)	27 (7.5)	10 (7.3)	0.480	1.440	0.796–2.605	0.227	0.921	0.436–1.945	0.829
Hypo-thyroidism	44 (11.8)	45 (12.5)	23 (16.8)	0.320	1.124	0.744–1.697	0.579	1.330	0.798–2.215	0.274

OR: odds ratio, CI: confidential interval, IVF: in vitro fertilization, BMI: body mass index, HTN: hypertension, GDM: gestational diabetes mellitus. data was shown as mean ± SD or number (%).

**Table 3 jpm-13-00444-t003:** Factors associated with abnormal high-density lipoprotein in postpartum.

	Normal	Borderline	Abnormal		Normal vs. Borderline to Abnormal			Normal to Borderline vs. Abnormal		
	(*n* = 385)	(*n* = 458)	(*n* = 26)	*P*	OR	CI	*p*	OR	CI	*p*
Age (years)	34.3 ± 3.8	35.1 ± 3.9	33.6 ± 4.7	0.011	1.037	0.997–1.078	0.071	0.884	0.786–0.994	0.039
Multi-parity	83 (21.6)	108 (23.6)	4 (15.4)	0.533	0.966	0.681–1.372	0.849	0.929	0.292–2.958	0.900
IVF	262 (31.9)	139 (30.3)	9 (34.6)	0.820	0.833	0.592–1.172	0.293	1.396	0.516–3.777	0.512
BMI (kg/m^2^)	20.3 ± 2.3	21.4 ± 3.2	22.4 ± 3.4	<0.001	1.182	1.116–1.252	<0.001	1.157	1.023–1.307	0.020
Weight gain (kg)	12.9 ± 4.1	13.8 ± 4.8	15.5 ± 5.4	0.002	1.085	1.042–1.131	<0.001	1.139	1.039–1.249	0.006
Weigh reduction (kg)	9.4 ± 2.9	9.4 ± 2.9	9.9 ± 3.3	0.372	0.941	0.882–1.005	0.068	0.880	0.756–1.024	0.098
Twin pregnancy	19 (4.9)	19 (4.1)	2 (7.7)	0.645	0.945	0.428–2.084	0.888	1.418	0.214–9.407	0.717
Preterm birth	37 (9.6)	36 (7.9)	3 (11.5)	0.587	0.780	0.442–1.378	0.392	0.876	0.178–4.322	0.871
HTN	5 (1.3)	19 (4.1)	5 (19.2)	<0.001	3.451	1.224–9.727	0.019	9.945	2.793–35.416	<0.001
GDM	24 (6.2)	29 (6.3)	4 (15.4)	0.183	0.878	0.481–1.604	0.672	3.116	0.914–10.619	0.069
Hypo- thyroidism	39 (10.1)	67 (14.6)	6 (23.1)	0.044	1.498	0.971–2.311	0.068	2.585	0.954–7.007	0.062

OR: odds ratio, CI: confidential interval, IVF: in vitro fertilization, BMI: body mass index, HTN: hypertension, GDM: gestational diabetes mellitus. data was shown as mean ± SD or number (%). Post hoc analysis—Age: normal vs. borderliine: *p* = 0.008, BMI: normal vs. borderline: *p* < 0.001; nomal vs. high risk: *p* < 0.001, Weight gain: normal vs. borderline: *p* = 0.006; normal vs. high risk: *p* = 0.006.

**Table 4 jpm-13-00444-t004:** Factors associated with abnormal triglyceride in postpartum.

	Normal	Borderline	Abnormal		Normal vs. Borderline to Abnormal			Normal to Borderline vs. Abnormal		
	(*n* = 730)	(*n* = 79)	(*n* = 60)	*P*	OR	CI	*p*	OR	CI	*p*
Age (years)	34.5 ± 3.9	35.5 ± 3.7	35.4 ± 3.9	0.028	1.045	0.991–1.103	0.107	1.066	0.989–1.149	0.096
Multi-parity	163 (22.3)	22 (27.8)	10 (16.7)	0.289	0.895	0.555–1.444	0.650	0.538	0.255–1.134	0.103
IVF	224 (30.7)	31 (39.2)	16 (26.7)	0.218	1.070	0.682–1.680	0.768	0.556	0.280–1.104	0.093
BMI (kg/m^2^)	20.7 ± 2.7	22.0 ± 3.4	22.3 ± 3.6	<0.001	1.178	1.106–1.254	<0.001	1.148	1.059–1.243	0.001
Weight gain (kg)	13.5 ± 4.5	13.6 ± 5.4	13.6 ± 4.0	0.945	1.108	1.054–1.165	<0.001	1.066	0.997–1.140	0.060
Weigh reduction (kg)	9.6 ± 2.9	8.4 ± 3.0	9.0 ± 3.0	0.001	0.791	0.727–0.861	<0.001	0.874	0.783–0.976	0.017
Twin pregnancy	32 (4.4)	5 (6.3)	3 (5.0)	0.727	2.447	0.888–6.743	0.084	3.499	0.799–15.317	0.096
Preterm birth	63 (8.6)	11 (13.9)	2 (3.3)	0.088	0.692	0.316–1.515	0.357	0.191	0.036–1.020	0.053
HTN	18 (2.5)	7 (8.9)	4 (6.7)	0.004	2.891	1.168–7.156	0.022	1.731	0.448–6.694	0.426
GDM	44 (6.0)	4 (5.1)	9 (15.3)	0.019	1.239	0.604–2.542	0.559	2.453	1.068–5.630	0.034
Hypo- thyroidism	90 (12.3)	12 (15.2)	10 (16.7)	0.512	1.165	0.679–1.998	0.579	1.278	0.608–2.689	0.518

OR: odds ratio, CI: confidential interval, IVF: in vitro fertilization, BMI: body mass index, HTN: hypertension, GDM: gestational diabetes mellitus. data was shown as mean ± SD or number (%). Post hoc analysis—Age: normal vs. borderliine: *p* = 0.029, BMI: normal vs. borderline: *p* < 0.001; nomal vs. high risk: *p* < 0.001, Weight reduction: normal vs. borderline: *p* = 0.001.

**Table 5 jpm-13-00444-t005:** Factors associated with abnormal low-density lipoprotein in postpartum.

	Normal	Borderline	Abnormal		Normal vs. Borderline to Abnormal			Normal to Borderline vs. Abnormal		
	(*n* = 474)	(*n* = 272)	(*n* = 123)	*P*	OR	CI	*p*	OR	CI	*p*
Age (years)	34.4 ± 3.8	35.0 ± 4.0	35.1 ± 3.9	0.092	1.029	0.990–1.068	0.146	1.024	0.970–1.081	0.382
Multi-parity	98 (20.7)	64 (23.5)	33 (26.8)	0.302	1.150	0.821–1.612	0.416	1.239	0.785–1.956	0.357
IVF	142 (30.0)	91 (33.5)	38 (30.9)	0.609	1.035	0.747–1.435	0.835	0.978	0.616–1.551	0.923
BMI (kg/m^2^)	20.8 ± 2.9	21.0 ± 2.9	21.3 ± 3.0	0.256	1.033	0.984–1.085	0.191	1.047	0.980–1.118	0.173
Weight gain (kg)	13.5 ± 4.6	13.6 ± 4.7	13.5 ± 4.0	0.931	1.022	0.985–1.061	0.250	0.997	0.946–1.051	0.914
Weigh reduction (kg)	9.5 ± 2.9	9.3 ± 3.0	9.5 ± 2.6	0.746	0.961	0.904–1.021	0.199	1.015	0.932–1.106	0.734
Twin pregnancy	20 (4.2)	16 (5.9)	4 (3.3)	0.431	1.264	0.597–2.677	0.541	0.708	0.215–2.332	0.570
Preterm birth	40 (8.4)	28 (10.3)	8 (6.5)	0.439	0.964	0.561–1.656	0.893	0.755	0.328–1.741	0.510
HTN	15 (3.2)	12 (4.4)	2 (1.6)	0.344	1.153	0.523–2.540	0.724	0.479	0.108–2.133	0.334
GDM	32 (6.8)	19 (7.0)	6 (4.9)	0.712	0.892	0.504–1.577	0.694	0.602	0.244–1.486	0.271
Hypo- thyroidism	53 (11.2)	37 (13.6)	22 (17.9)	0.129	1.310	0.874–1.962	0.191	1.492	0.887–2.510	0.132

OR: odds ratio, CI: confidential interval, IVF: in vitro fertilization, BMI: body mass index, HTN: hypertension, GDM: gestational diabetes mellitus. data was shown as mean ± SD or number (%).

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
