# Peer review of "Intrapartum Factors Affecting Abnormal Lipid Profiles in Early Postpartum Period"

_jpm, 2023, doi:10.3390/jpm13030444_

Round 1

Reviewer 1 Report

The study is of interest as most previous studies have not included nonpregnant lipid values.

In line 49 you use the Belo study to illustrate that abnormal levels of lipid early in pregnancy are associated with an increased risk of PE. This study only looked at lipid levels in late pregnancy and the puerperium so another reference will need to be included.

The English needs to be improved eg I suggest changing the table titles to "Factors associated with ....." There are many spots where English should be improved

Lines 276 to 278- it needs to be mentioned that the abnormal lipid profiles postpartum are likely factors (or associated with other factors such as obesity) which predisposed the participants to the development of preeclampsia and GDM so the HTN and GDM are not independent factors

Reviewer 2 Report

1. Introduction

I've appreciated the coincise introduction. I'd suggest to widen the background of the study, remarking all the conditions related to obesity and abnormal lipid profiles during pregnancy and postpartum period (including post partum hemorrhage, C-section complication, placental defects and fetal growth restriction). 

2. Material and methods: 

Line 70: Please remark the study design (retrospective cohort).

Line 70-75: Please add a schematic diagram of the screening and inclusion process.

Line 82: Please provide the date of approval.

Lines 85-96: Please provide references for criteria applied.

Line 93: Please specify the type of DM. I believe it's DM type 2, due to uncommon lipid profiles changes in DM type 1.

Line 96: Please specify type of medications for any comorbidity investigated and whether those determined a complete control of the condition itself (i.e., euthyroidism or normal serum glucose level).

Lines 97-107: References are needed.

Lines 97-115: I believe that a longitudinal model would have been more appropriate to investigate gestational changes in lipids' profiles. If a single time point was taught to be suitable for comparisons, please indicate the reason of your choice and how did you incorporate those changes in your evaluation.

3. Discussion:

Please expand your references, providing a larger overview of potential molecular mechanisms involved. As an example, to justify the association between life-style factors, obesity and PE/FGR, several epigenetic changes might be able to explain the findings. Please add these references to your discussion (DOI:10.1007/s40291-022-00611-4DOI: 10.1093/humupd/dmz025; DOI: 10.1007/s00125-019-4951-9) and expand this argumentation.

Reviewer 3 Report

The authors describe intrapartum factors affecting abnormal lipid profiles in the early postpartum period which were retrospectively analyzed. The manuscript is well written, but would profit from a language editing. The content is relevant and of interest to the reader, I therefor recommend  to accept the manuscript after minor revisions. The manuscript is clearly structured and easy to read, the conclusions are consistent with the detailed results and arguments. The main question is address by the results and the authors contribute to the field with new facts. 
